# Intestinal effect of faba bean fractions in WD-fed mice treated with low dose of DSS

**Dimitrios Papoutsis[1], Sérgio Domingos Cardoso Rocha[2], Anne Mari Herfindal[1], Siv Kjølsrud Bøhn[1], Harald Carlsen[1]** *

**1** Faculty of Chemistry, Biotechnology and Food Science, Norwegian University of Life Sciences, Ås, Norway, **2** Department of Animal and Aquacultural Sciences, Faculty of Biosciences, Norwegian University of Life Sciences, Ås, Norway

* harald.carlsen@nmbu.no

## Abstract

Rodent studies have shown that legumes can reduce chemical induced colonic inflammation, but the role of faba bean fractions for colon health has not been described. We have investigated the role of protein and fiber fractions of faba beans for colonic health and microbiota composition in a low-grade inflammation mice-model when incorporated in a Western diet (WD). The diet of sixty C57BL/6JRj male mice was standardized to a WD (41% fat, 43% carbohydrates) before were randomly assigned to four groups (n = 12) receiving either 1) WD with 30% of the protein replaced with faba-bean proteins, 2) WD with 7% of the fiber replaced with faba-bean fibers, 3) WD with protein and fiber fractions or 4) plain WD (n = 24). Low-grade inflammation was induced by 1% dextran sodium sulfate (DSS) given to mice for the last six days of the trial. Half (n = 12) in group 4) were given only water (controls). Prior to DSS, body weight, energy intake, glucose and insulin tolerance assays were performed. Inflammatory status in the colon was assessed by biomarkers of inflammation and qRT-PCR analyses of inflammatory related genes. Fecal microbiota composition was assessed by 16S rRNA gene sequencing. 1% DSS treatment increased levels in fecal lipocalin-2 and induced disease activity index score, but the presence of faba bean fractions in WD did not influence these indicators nor the expression level of inflammatory associated genes. However, the mice that had faba-bean proteins had a lower amount of Proteobacteria compared the group on plain WD. The Actinobacteria abundance was also lower in the group that had fiber fraction from faba-beans. Overall, outcomes indicated that in a low-grade inflammation model, replacement of protein and or fiber in a WD with faba bean fractions had marginal effects on inflammatory parameters and colonic microbiota.

## Introduction

Legumes and pulses (the dry seed of the legume), which belong to the Fabaceae family, constitute an inexpensive food source with high nutritional value, often attributed to their richness in proteins (20–35%), dietary fibers and various phytochemicals such as polyphenols. The World Health Organization has actively encouraged to include more legumes in the daily meal due to their high nutritional value [1–3].

**Data Availability Statement:** All relevant data are within the pass and its Supporting Information files

**Funding:** DP: PhD scholarship fundend by Norwegian University of Life Sciences The study was a part of FoodProFuture, funded by Norwegian

Research Council (https://www.forskningsradet.no/) Grant#267858.

**Competing interests:** The authors have declared that no competing interests exist.

Data from epidemiological studies have shown that legume consumption is positively associated with improved blood cholesterol profile [4], reduced risk of cardiovascular diseases [5, 6], type-2 diabetes [6, 7], metabolic syndrome [8–10] and colorectal cancer [11, 12]. In addition, feeding trials in rodents with different types of legumes have demonstrated a positive impact against colonic inflammation, chemically induced by dextran sodium sulfate (DSS) [13–15].

The beneficial health effects of legumes are probably multicausal and it is unlikely that one single nutrient is responsible. Dietary fibers, which increase the bulk of stool, maintain regular bowel movements and are sources of short chain fatty acids from bacterial fermentation, can prevent the advent of inflammation and several chronic diseases [16, 17]. Furthermore, plant proteins are proposed to contribute more on overall health state than animal proteins [18]. According to some studies, high intake of animal proteins, particularly from red meat, is linked to inflammatory bowel diseases (IBDs) [19], whereas the risk is lower with increased consumption of plant proteins [20]. These positive indications may be related to differences in amino acid profiles [21], or the presence of bioactive peptides arising during the digestion of plant proteins [22, 23]. A mouse study with pea albumin extracts in the diet demonstrated reduced inflammation and differences in microbiota composition after DSS treatment. The authors suggested that these effects were attributed to certain bioactive protein components called Bowman Birk inhibitors, present in peas and other legumes [24].

The presence of antinutrients in the legume protein package, such as lectins, saponins, and enzyme inhibitors, may be responsible for adverse health effects and often constitutes a major concern. However, through processing (soaking, cooking or other thermal treatments) the concentration of antinutritional factors can be significantly reduced [25]. Moreover, the presence of antinutrients in moderate amounts help to reduce blood glucose, plasma cholesterol, triglycerides an even reduce cancer risk [26–28].

Faba bean (*Vicia faba L*), which is also referred as broad bean, field bean and horse bean [29], represents a popular dish mainly in Middle East and Mediterranean region [30]. Faba beans is the fourth most cultivated legume after peas, chickpeas and lentils (FAOSTAT, 2019). Faba beans can tolerate lower temperature [31], allowing growth in different climate zones, including the Nordic countries [32]. Notably, not only levels of nutrients and antinutrients among various cultivars of faba beans can vary significantly [33] but also the amount of soluble fiber in different forms of faba bean protein ingredients (protein-rich flour, protein isolate) [34].

So far very few studies on potential health effects of faba beans have been carried out. We therefore performed a feeding trial with high fat Western-like diet (WD), where we incorporated fractions from faba beans (protein, fiber or both fractions together). The aim was to investigate colon health status under a WD and low-grade inflammation induced by a low concentration of DSS (1%) and we hypothesized that adding cooked fractions of faba beans in a WD would manifest a lower degree of intestinal inflammation (after 1% DSS treatment).

Prior to DSS treatment, the WD with protein fraction significantly increased body weight gain of mice ($P<0.05$) but not glucose and insulin tolerances compared to those fed the WD without faba bean fractions. During 1% DSS treatment in drinking water, evaluation of colon through inflammatory markers (LCN2, LBP) showed no significant effects in mice following WDs with faba bean fractions compared to WD-fed mice. Evaluation of microbiota composition between the different dietary groups demonstrated minor shifts from phyla level to genera level.

## Materials and methods

### Animals and diets

Six-week-old male C57BL/6JRj mice (n = 60) were purchased from JANVIER LABS (Le Genest-Saint-Isle, France) and housed in individually ventilated cages (Innorack, Innovive, San

Diego, CA; n = 4 per cage) under controlled conditions (12 hours light-dark cycle; 23±2˚C, 45–55% humidity). Upon arrival, mice were acclimatized for two weeks and fed standard rodent chow diet (RM1, SDS Diet, Essex, UK). After acclimatization, all mice switched to a Western Diet (WD) (D12079B, 43.6% CHO—41.2% fat) for eight weeks before they were allocated to four dietary groups (for another eight weeks), in which fiber, protein or both fractions from faba beans (type Vertigo, Denmark) were included. Those groups were i) a WD with no supplement (WD, n = 24) ii) a WD supplemented with 30% faba bean protein fraction (WD +PF, n = 12), iii) a WD supplemented with 7% of faba bean fiber fraction (WD+FF, n = 12) and iv) a WD supplemented with both faba bean protein (30%) and fiber fraction (7%) (WD +BF, n = 12). The faba bean strain, Vertigo was grown and harvested in Denmark and shipped to Norway for further processing. The protein fraction was obtained by dry milling and air-classification (Skjelfoss Korn AS, Hobøl, Norway) of faba beans after the hull removal (hull fraction). The fiber fraction was acquired from the hull fraction. The hull fraction was milled using a Retsch ZM 100 mill (Retsch Gmbh, Haan, Germany) comprising a 0.5mm sieve before further use in the mice feeds.

Faba bean fractions were cooked (10 min) and freeze-dried before being shipped to Research Diets (Madison, WI) for final preparation of the mouse diets. The reason for cooking them was to reduce the concentration of toxic antinutrients such as vicine and convicine, which often may cause in humans and animals a hemolytic disease called favism [35, 36]. Composition of the diets is presented in S1A and S1B Table.

Following the six-week feeding with the WD +/- faba bean fractions, 1% DSS was introduced in the drinking water of mice (n = 12/group) for six days to induce low-grade colon inflammation. To score disease activity from the 1% DSS treatment, mice were evaluated every two days to determine a disease activity (DAI) index score, which considered body weight, activity level, posture and stool quality. Half of the WD-fed mice (n = 12) were given water only (without DSS) and used as negative controls. The experimental timeline is presented in Fig 1.

Water and food were provided *ad libitum*. Body weights and food intake were recorded once per week. Experimental procedures were approved by the Norwegian Animal Research Authority (Mattilsynet, FOTS ID 14805) in accordance with the Federation of European Laboratory Animal Science Associations (FELASA).

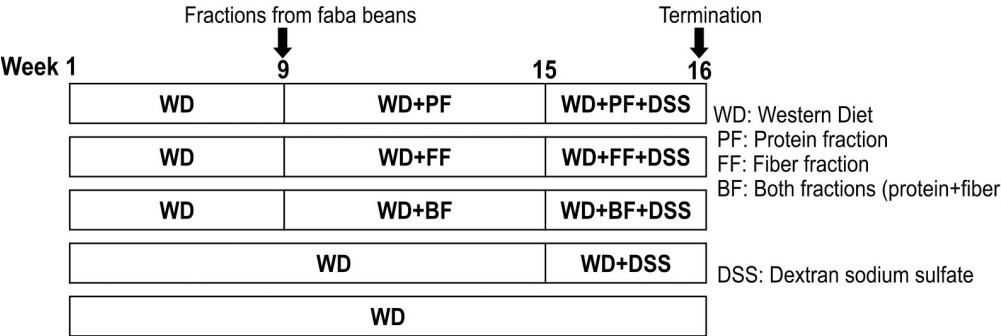

**Fig 1. Timeline of feeding trial prior and during 1% DSS treatment.** For eight weeks, all C57BL/6JRj mice (n = 60) were fed a WD. From week 9 mice (n = 12) were divided into four dietary groups, in which faba bean fractions were introduced in the WD: 1) WD incorporated with 30% faba-bean protein fraction (WD+PF), 2) WD with 7% fiber fraction (WD+FF), 3) WD with both protein and fiber fraction (WD+BF) or 4) WD without faba bean content (n = 24). The mice were followed for another eight weeks. The last six days before experiment termination, mice were exposed to 1% DSS in the drinking water expect half of the WD only group (n = 12), serving as negative controls for the DSS effect.

## Glucose tolerance test (OGTT) and insulin tolerance test (ITT)

In the third and fourth week of the feeding trial with the different diets (WD, WD+PF, WD +FF, WD+BF), assays for glucose homeostasis and insulin sensitivity were carried out, respectively. For the oral glucose tolerance test (OGTT), animals were fasted for 6 hours and a fixed dose of D-glucose (Sigma-Aldrich, 2 g/kg, meaning 300 μl of glucose solution 20% per mouse) was administered *per os* and blood (~3–5 μL) was collected from tail at different time points. Blood glucose levels were measured by a glucometer (Accu-Check Ⓡ, Roche Diagnostics) at baseline and 15, 30, 60 and 120 minutes after glucose administration. For the insulin tolerance test, mice fasted for 4 hours before human insulin (I2643, Sigma-Aldrich) was injected i.p. (0.25 U/kg). Blood glucose was measured at the baseline and 30, 60, 120 minutes after insulin injection.

## Sampling and histology

Samples were collected on day six of 1% DSS treatment (termination of experiment). Mice were anesthetized with a cocktail containing Zoletil Forte (Virbac, Carros, France), Rompun (Bayer, Oslo, Norway), and Fentadon (Eurovet Animal Health, Bladel, The Netherlands) (ZRF; i.p. 0.1 mL ZRF/10 g body weight), with the following active ingredients: Zolezepam (32 mg/kg), Tiletamine (32 mg/kg), Xylazine (4.5 mg/kg), and Fentanyl (26 mg/kg). Prior to mice euthanasia by cervical dislocation, blood (0.5–1 mL) was collected by cardiac puncture into syringes with ~50 μL NaEDTA (50 mM) as anticoagulant. To obtain plasma blood was centrifuged at 6,000xg for 10 min at 4˚C. After opening the colon longitudinally, mucosa was collected by gentle scraping with a sharp glass slide. Colonic mucosal samples were kept in RNAlater (Sigma-Aldrich) to preserve RNA. Fecal pellets were collected from the colon. All samples were stored in -80˚C.

Sections from the distal colon were fixed following the protocol already described [37]. Generally, the colon lumen was washed with modified Bouin's fixative (50% ethanol, 5% acetic acid in $dH_2O$), opened longitudinally to expose the luminal side and then wrapped around a toothpick with the luminal side facing inwards. Samples were then kept in 10% buffered formalin overnight at room temperature and transferred to 70% ethanol and stored at 4˚C until samples were embedded in paraffin according to protocol provided by the university imaging core facility. The paraffin embedded samples were cut in 7μm thick sections and stained with hematoxylin and eosin before they were judged for structural damage under a light microscope (DM6B, Leica, Germany).

## RNA extraction and qRT-PCR

RNA from colonic mucosa samples collected on day six of the DSS treatment (termination ay) was extracted with the NucleoSpin RNA/Protein Purification kit (Macherye-Nagel, Düren, Germany). Because DSS reduces efficiency of both reverse transcriptase and PCR reactions, all RNA samples were purified using lithium chloride (LiCl) described by Viennois *et al.* [38].

cDNA synthesis from RNA was performed (S3 Table) using iScript cDNA Synthesis Kit (1708891, Bio Rad), whereas FirePol EvaGreen qPCR Supermix (08-36-00001, Solis BioDyne) was used for the qRT-PCR reaction in a Light Cycler 480 Instrument II (Roche). The parameter settings were: 12 min at 95˚C; 40 cycles of 15 sec at 95˚C followed by 20 sec at optimized primer annealing temperature; 20 sec at 72˚C (S4 Table). LinRegPCR Software (2017.1.0.0, Amsterdam UMC) was used to calculate Cq values of the colon samples and each primer efficiency. Primers used for mRNA expression (Thermo Fisher Scientific) and are presented in S5 Table.

## LBP measurement

ELISA assay for determination of lipopolysaccharide binding protein (LBP) was performed in plasma samples collected on day six of the DSS treatment (termination day) according to the manufacturer (Biometec, Greifswald-Rostock, Germany). Plasma samples from control mice were diluted 800 times (as recommended in the guidelines), whereas those from 1% DSS-treated mice were diluted from 1,200 to 1,600 times. Optical density of each sample at 450nm was measured with a spectrophotometer (SpectraMax M2, Molecular Devices) and concentration of the protein estimated based on standard curve using 4-parameter logistic curve fit.

## Lipocalin-2 measurement

Mouse Lipocalin-2/NGAL DuoSet ELISA from Research and Development systems (R&D systems, USA) was used for measuring lipocalin-2 protein in extracts from fecal samples collected on day six (termination day) of DSS treatment. Fecal extracts were made from feces samples (weighed and stored at -80C) reconstituted in PBS containing 0,1% Tween 20 (1ml buffer per 100mg feces). The samples were vortexed for 20min to obtain a homogenous fecal suspension. The supernatant was collected from each sample after centrifuging the samples for 10min at 12.000 rpm (4˚C), as described by Chassaing *et al*. [39]. Prior to the assay, fecal samples from control mice and 1% DSS was diluted 20 and 20.000 times respectively. The concentration was measured by optical density as described for LBP measurements.

## Microbiota analysis

**DNA extraction from feces and library preparation.** Right after dissection, fecal samples were collected from the colon and placed in 400 μL S.T.A.R buffer (Roche, USA) with acid-washed glass lysing beads (approximately 0.2g <106 μm, 0.2g of 425–600 μm and 2–4 beads of 2mm Sigma-Aldrich) and stored at -80˚C for further processing. All samples were homogenized twice on FastPrep 96 (1,800 rpm, 40 sec, 5 min cooling step in-between, MP BioMedicals). Processed samples were then centrifuged (13,000 rpm, 10 min) and 50 μL supernatant was transferred to 96-well plates for protease treatment and DNA extraction using Mag Mini LGC kit (LGC Genomics, UK) on KingFisher Flex DNA extraction robot (Thermo Fisher Scientific, USA) according to the manufacturer's instructions.

**Gene sequencing of 16S Rrna.** After DNA extraction, 16S rRNA gene was amplified by PCR using prokaryote-targeting primers against the variable region V3-V4 with an amplicon length of 466bp. PCR primers for amplification and conditions for library preparation are presented in the S6 and S7 Tables. As DSS in fecal samples has an inhibitory effect on the PCR reaction (identified through dilution series on qPCR), we diluted the extracted DNA from feces 1:4 prior to amplicon PCR (total dilution of 1:100 in the PCR reaction). PCR product (466 bp) was purified with AMPure XP (Beckman-Coulter) and 10 further PCR cycles with index primers were performed (S8 and S9 Tables) resulting in PCR product of approximately 594 bp. The sequences of primers in index PCR are shown in S10 Table. All PCR products were qualitatively confirmed by electrophoresis on 1.5% agarose gel. Quantification of DNA concentrations of index PCR products, normalization and pooling of these index PCR products were followed by purification of the pooled library with Sera Mag Beads by following the AMPure XP protocol. The pooled library was diluted to 6 pM and sequenced with the MiSeq Reagent Kit V3 (cat. nr. MS-102-3003) on the Illumina MiSeq following Illumina's protocol (16S Metagenomic Sequencing Library Preparation Part# 15044223 Rev. B), except we used nuclease free-water instead of Tris for PhiX library dilution. 20% PhiX served as an internal control.

**Assigning taxonomy.** Resulting 300 bp paired-end reads from MiSeq platform were filtered using QIIME and OTU (Operationally Taxonomic Unit) clustered based on 97% identity using closed-reference OTU *usearch* algorithm (version 8) [40, 41] against SILVA database (version 128) [42]. 6,500 sequences per sample were chosen as a cut-off to normalize (rarefy) the sequencing data.

**Statistical analysis.** Statistical analyses were performed using GraphPad Prism (version 8.3.1 for Windows, GraphPad Software, San Diego, CA). Data are presented as individual values with group means ± standard error of the mean (SEM). Normal distribution was tested using the Shapiro-Wilk normality test. Based on whether normal distribution was achieved or not, parametric and non-parametric models were used respectively. *P*-values smaller than 0.05 were considered significant.

Prior to DSS treatment, body weight, energy intake and insulin tolerance were analyzed by the mixed effect model whereas glucose tolerance by repeated measures 2-way ANOVA. The latter statistical method was used for analyzing the body weight change (%) of the four diet groups during the six days of DSS treatment whereas one-way ANOVA followed by Tukey's post-hoc analysis and the non-parametric test Kruskal-Wallis with Dunn's multiple comparisons for colon length and DAI on day six of 1% DSS treatment respectively. For the statistical analysis of proinflammatory cytokines and biomarkers of inflammation, the control mice group of mice was excluded and either one-way ANOVA followed by Tukey's post-hoc analysis or Kruskal-Wallis with Dunn's multiple comparisons was applied. Analysis of beta-diversity was conducted in R (version 4.0.0). Weighted UniFrac distances were calculated using QIIME default scripts (core_-diversity_analyses.py) and are based on the normalized (rarefied) OUT table. Non-metric multi-dimensional scaling (NMDS) of weighted UniFrac distances was performed using the metaMDS function from the *vegan* package [43] with autotransform = FALSE and try = 100. Global PERMANOVA on weighted UniFrac distances was performed using the adonis function from the *vegan* package with 999 permutations. Pairwise PERMANOVA was performed by applying the pairwise.perm.manova function from the *RVAideMemoire* package [44].

For linear discriminant analysis effect size (LEfSe), relative abundances of taxa were used [45]. To identify statistical differences, factorial Kruskal-Wallis sum test, followed by pairwise Wilcoxon test, both with an alpha value of 0.05, were used. The threshold on the linear discriminant analysis (LDA) score was set at 2.0 with a multi-class analysis against all.

## Results

### Body weight, energy intake and glucose regulation before DSS treatment

Prior to DSS treatment the eight-week impact of the different diets was examined. Mice fed a WD+PF gained significantly more weight (*P*<0.0001) compared to mice given WD or WD+FF. No difference in weight gain was observed between WD and WD+FF fed mice (Fig 2A). The energy intake *per* mouse was significantly higher in mice fed WD+PF compared to WD+FF fed mice (*P* = 0.0148) (Fig 2B), but no difference was found between any of the other groups. In terms of glucose regulation, no significant differences were observed between the dietary groups judged by the glucose tolerance test (*P* = 0.8037) or the insulin tolerance test (*P* = 0.1269) (Fig 2C and 2D).

### Impact of faba-bean fractions on body weight and colon during 1% DSS treatment

Before evaluating the impact of the faba-bean fractions initial investigations were performed to assess the low-inflammation model induced by 1% DSS treatment. During the first four

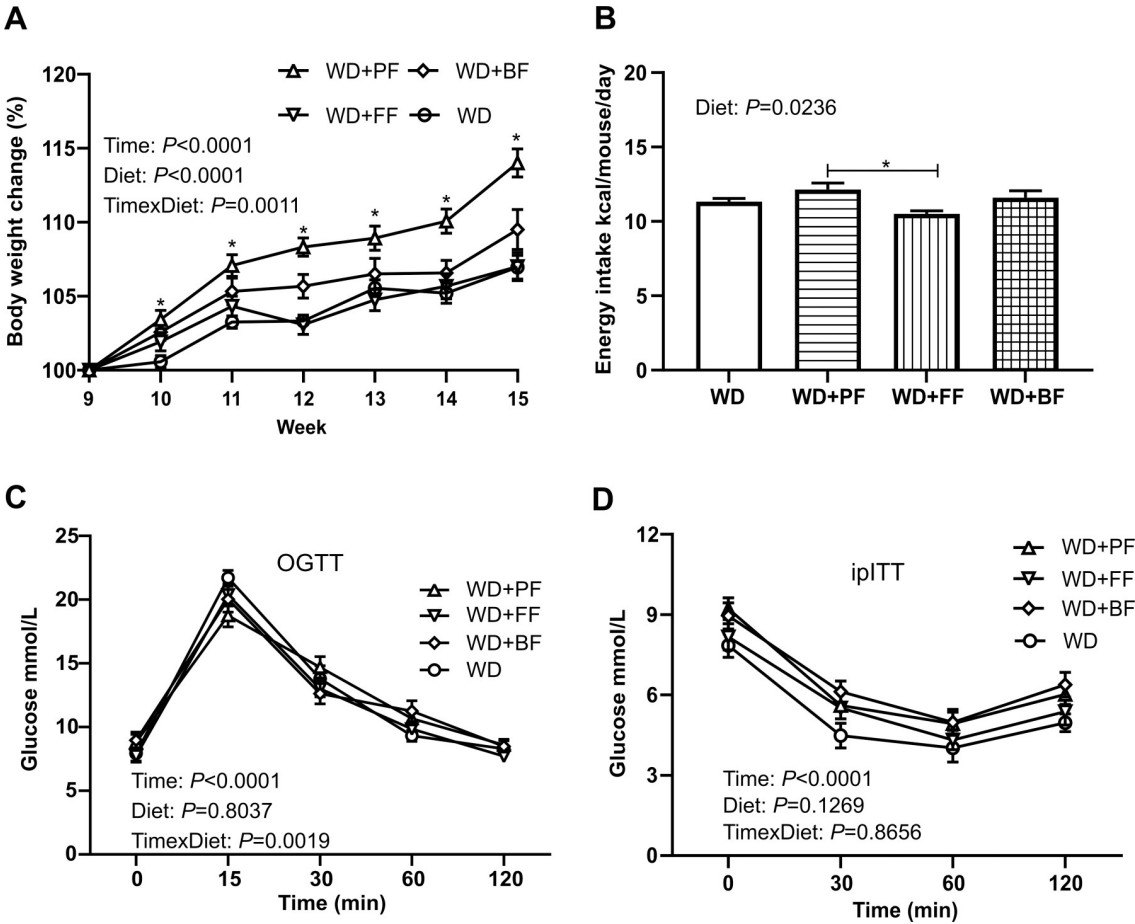

**Fig 2. Bodyweight development, energy intake and glucose regulation prior to DSS treatment in WD fed C57BL/6JRj mice.**
Bodyweight development (**2A**) during the feeding trial with modified WDs (n = 12 except WD-fed mice, n = 24). Estimated energy intake (**2B**) per mouse per day from week 9 to week 15. OGTT (**2C**) after 4 hours of fasting at week 3 (n = 12). IpITT (**2D**) after 6 hours of fasting at week 4 (n = 12). * means significantly different ($P < 0.05$). Values are means ± SEM. For panels A, B, D mixed-effect model whereas for panel C 2-way ANOVA with repeated measures was applied.

days of 1% DSS administration, no visible signs of disease were observed in any of the mice i.e., the mice continued to increase in weight due to WD (Fig 3A). At day six the weight increase had ceased in all diet groups and even mildly decreased from week 4. Also, mild signs of disease were observed in the majority of the mice judged by the animal researcher, visually inspecting stool quality, body posture and activity levels summarizing the DAI score. At termination day, colon length of all mice treated with 1% DSS was shorter than WD control mice, but no significant differences were found in terms of colon length between diet groups ($P = 0.1556$) (Fig 3B). Furthermore, histological staining of colon tissue for 1% DSS treated mice revealed no structural damage in colonic mucosal tissue (S1 Fig). Also, neither of the faba bean fractions impacted the overall DAI scores ($P = 0.6768$) on day six (termination day) (Fig 3C).

## Effect of diet on gene expression and biomarkers of inflammation with 1% DSS

We next assessed mRNA expression levels of the inflammatory related genes; *Tnf*-a, *Il-1b*, *Il-6*, *Nox2*, *Nos2* and *Ptgs2* in colonic mucosa. While mild signs of disease were observed as

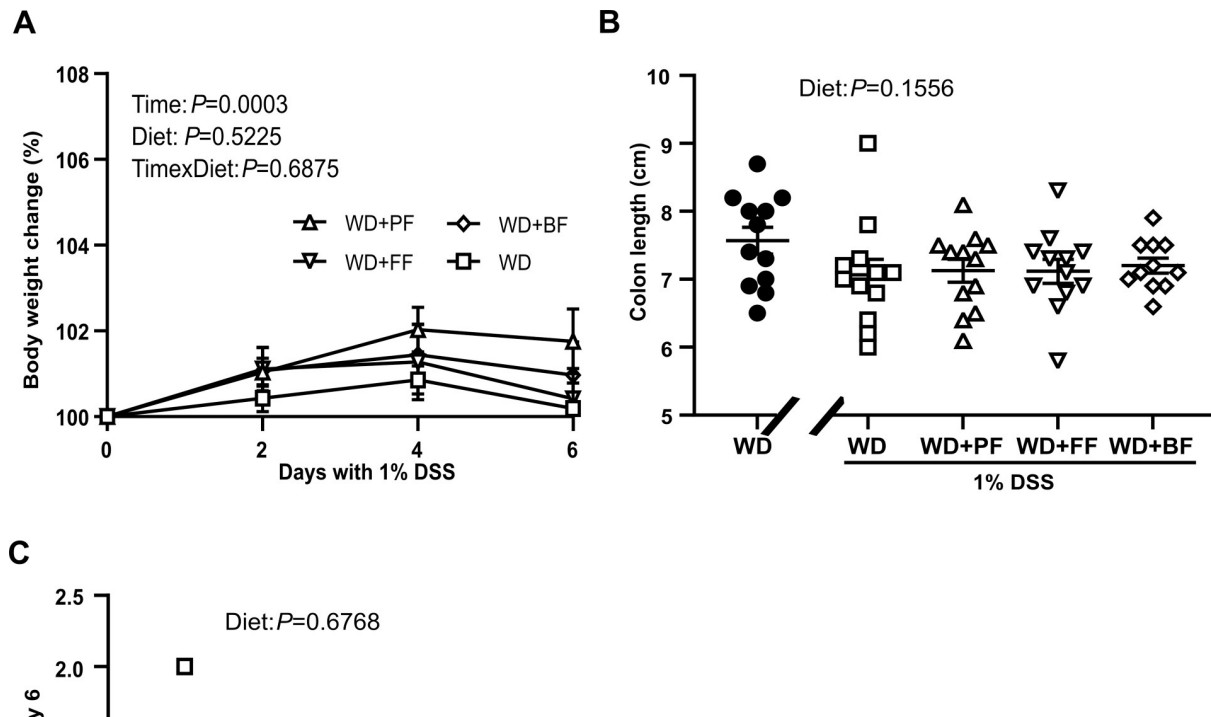

**Fig 3. Bodyweight (%) development during 1% DSS in WD fed C57BL/6JRj mice during 6-days treatment.** Change in bodyweight (%) measured on day 0, 2, 4 and 6 (**3A**). Colon length in cm (**3B**). Control mice indicated by black circles in the figure. For panel A 2-way ANOVA was used whereas for panels B and C one-way ANOVA and Kruskal Wallis test, respectively. Values are mean ± SEM. DAI score on day six of 1% DSS treatment (**3C**).

described in the previous section the 1% DSS treatment did not lead to induced expression of these genes when compared to the negative controls that were not exposed to DSS. Furthermore, no impact of the faba bean fractions was found on the gene expression (Fig 4A–4F). In contrast, the fecal levels of LCN2 were robustly upregulated by 1% DSS ($P = 0.0431$), but apart from WD+PF vs WD+BF ($P = 0.0286$) no differences in LCN2 levels were observed between the faba bean fraction groups (Fig 4G). Also, even as the plasma levels of LBP were overall higher in 1% DSS exposed mice compared to the negative WD-control mice, no difference could be observed between the faba-bean fraction groups ($P = 0.2217$) (Fig 4H).

## The presence of faba bean fractions in WD and during 1% DSS treatment brought shifts in microbiota profile

Furthermore, we investigated if replacement of protein and fiber content in the WD with different faba bean fractions could induce microbial shifts in colonic content. Alpha-diversity in

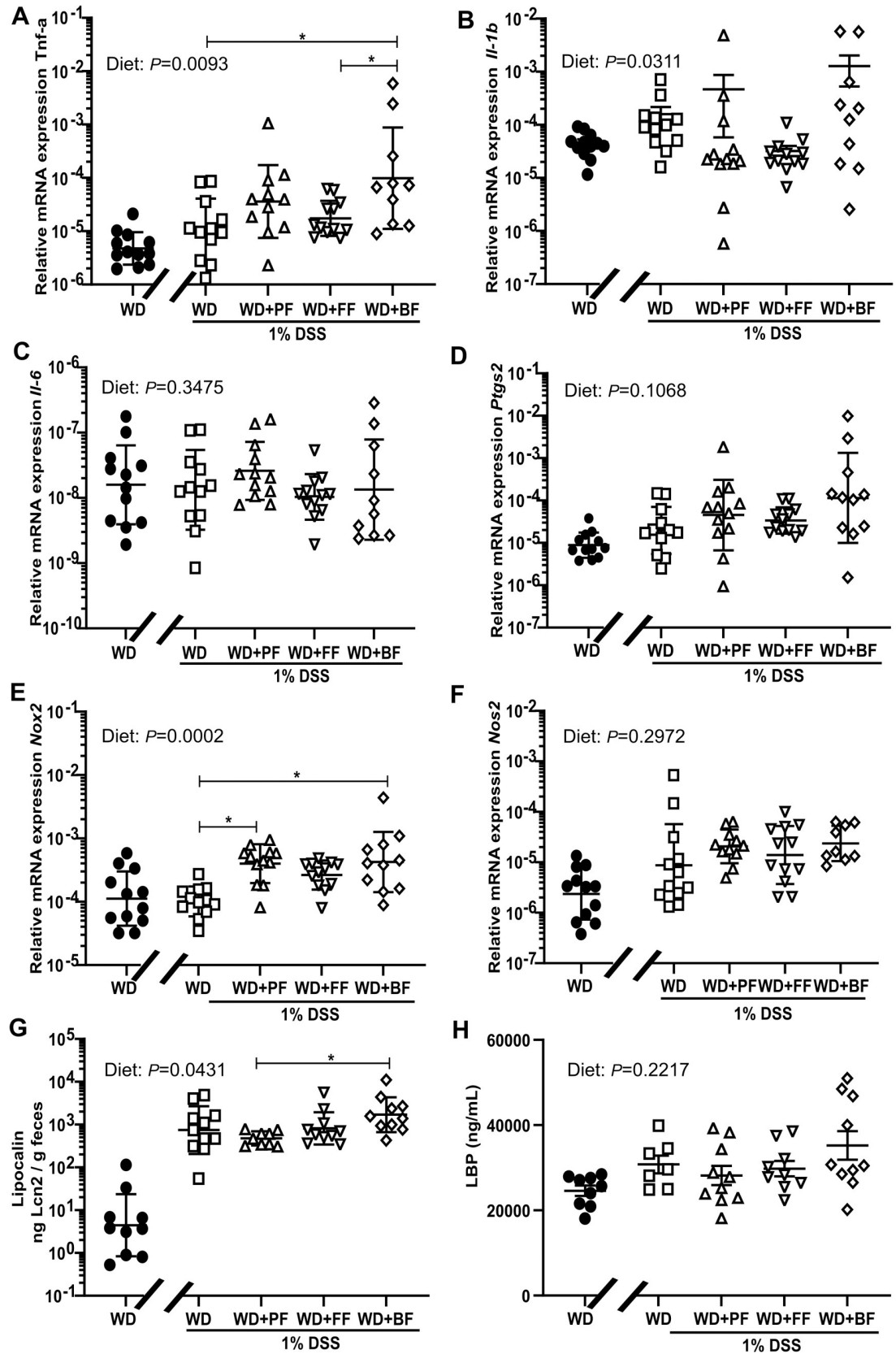

**Fig 4. Gene expression and inflammatory markers.** Impact of faba bean fractions on the expression of proinflammatory (*Tnf-a* (**4A**), *Il-1b* (**4B**), *Il-6* (**4C**), *Ptgs2* (**4D**)), and ROS associated genes (*Nox2* (**4E**), *Nos2* (**4F**)) in mucosa from the proximal colon (n = 10–12). Lipocalin-3 concentration in feces (**4G**) and LBP levels (n = 9–11) (**4H**) in plasma of WD fed C57BL/6JRj mice after 1% DSS treatment. WD+PF: WD incorporated with 30% faba-bean protein fraction; WD+FF; WD with 7% fiber fraction, WD +BF; WD with both protein- and fiber fraction, WD without faba bean content (n = 24). Values are mean ± SEM. One-way ANOVA with Tukey's multiple comparison for analysis in panels A-H, except panels B and E, where Kruskal-Wallis with Dunn's multiple comparisons test was used. Control mice indicated by black circles in the figures.

fecal samples was assessed by using the Shannon index on 16S rRNA sequencing data. No impact of DSS was found on the alpha-diversity when comparing the DSS exposed mice to the negative controls. However, among the DSS treated groups a significant difference was found between the WD and the WD+PF ($P = 0.0104$) while no significance was observed between the remaining groups (Fig 5A). Moreover, beta-diversity by using non-metric multidimensional scaling (NMDS) of weighted UniFrac distances, was examined giving an overall $P = 0.001$ between dietary groups. The DSS treatment was the main factor for group dispersion with an overlap between the clusters of groups exposed to 1% DSS indicating no impact of the faba-bean fractions on the beta-diversity (Fig 5B).

When examining differences in taxonomy at the phyla level, differences in the relative abundances were found between the groups (Fig 5C). Firstly, the negative control had a significantly higher level of Firmicutes and lower level of Bacteroidetes as visualized in Fig 5D showing the Firmicutes/Bacteroidetes ratio (Fig 5D). When comparing the diet groups that were exposed to DSS (one-way ANOVA), the relative abundance of Actinobacteria was significantly higher in WD ($P = 0.0130$) and WD+PF mice ($P = 0.0255$) compared to the WD+FF mice (Fig 5E). In addition, all DSS treated WD mice had higher relative abundance levels of Proteobacteria and Verrucomicrobia compared to the WD control mice (Fig 5F and 5G). Interestingly, DSS and fed with faba bean fractions had lower relative abundance levels of those two phyla compared to DSS-treated mice fed solely a WD. Although statistical significance was only found in the WD+PF fed mice ($P = 0.04$), the tendency for changes in the microbiota profile at phylum level was evident when faba bean fractions were present in WD.

To get a deeper understanding on the taxonomic levels, we applied the linear discriminant analysis effect size (LEfSe), which can identify differences in microbial communities based on relative abundances and hence allow us to statistically test differences among the experimental groups. When comparing the WD group and the diet groups that had replaced protein and or fiber content with faba-bean fractions several bacterial clades were detected to be different (Fig 6A). Genera of *Bifodobacterium* (Fig 6B), *Alloprevotella* (Fig 6C) together with *Prevotellaceae UCG-001* (Fig 6F) and *Enterorhabdus* (Fig 6G) were indicated in WD+PF, WD+FF, WD+BF respectively (Fig 6B). In WD+DSS fed-mice *Tyzzerella* (Fig 6D) and *Ruminococcaceae UCG-005* (Fig 6E) showed significantly high relative abundance ($P<0.05$) compared to the faba-bean fraction groups.

It is worth mentioning that *Bifidobacteria* in mice consuming WD+PF was similar to WD-fed mice without DSS treatment (as illustrated earlier in Fig 5A), while *Alloprevotella* and *Prevotellaceae UCG-001* manifested higher relative abundance in mice eating a WD supplemented with faba bean fractions (mainly fiber fraction) than mice following a WD with or without 1% DSS treatment.

## Discussion

In the present study we examined the effect of a WD supplemented with two different fractions from faba beans in mice treated with 1% DSS for six days. Contrary to commonly used 2–5% DSS concentration for inducing colitis, we used 1% DSS to cause an irritation of the colon

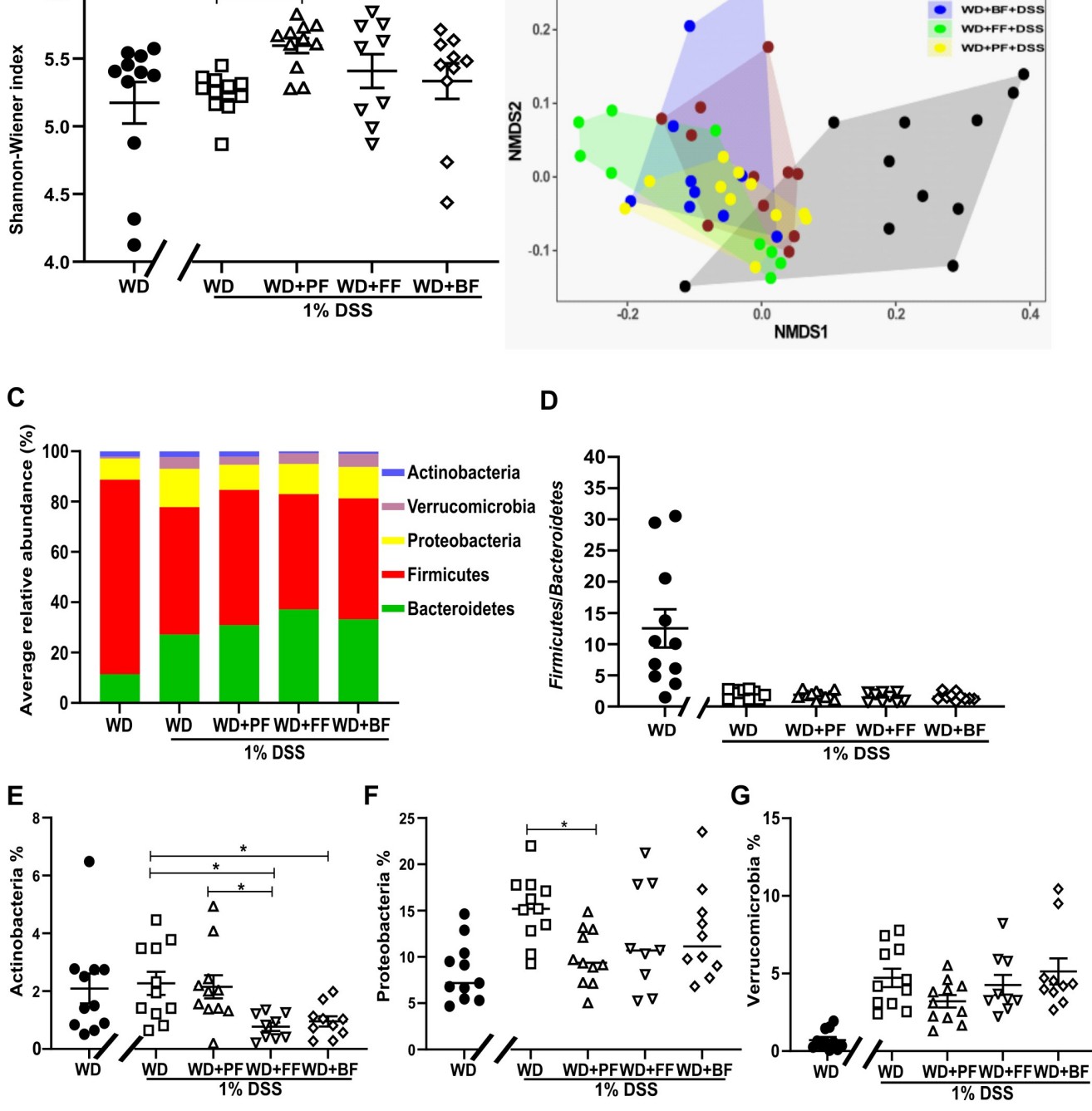

**Fig 5. Impact of various faba-bean fractions with WD (WD-PF, WD-FF and WD-BF) on microbiota measures in C57BL/6JRj exposed to 1% DSS.**
WD without DSS treatment served as negative control. Alpha-diversity (Shannon index) of colonic bacterial communities in feces from WD mice represented as mean with SEM n = 8–11 in each dietary group (**5A**). Groups were compared using Kruskal-Wallis test since one-way ANOVA could not be performed due to violation of normality assumption. Impact of faba bean fractions on beta-diversity was explored using non-metric multidimensional scaling (NMDS) of weighted UniFrac distances between fecal samples (**5B**). Colors indicate dietary group. *P*-value in the plot from global PERMANOVA. Average relative abundance for all detected phyla for each group in fecal samples (**5C**). Firmicutes/Bacteroidetes ratio in feces (**5D**), relative abundance of Actinobacteria (**5E**), Proteobacteria (**5F**) and Verrucomicrobia (**5G**) phyla. For panel D Kruskal-Wallis test with Dunn's multiple comparisons whereas for panel E-G one-way ANOVA with Tukey's multiple comparison was conducted. * means significantly different (*P*<0.05). Values are expressed as means ± SEM, n = 9–11. Control mice indicated by black circles in the figures.

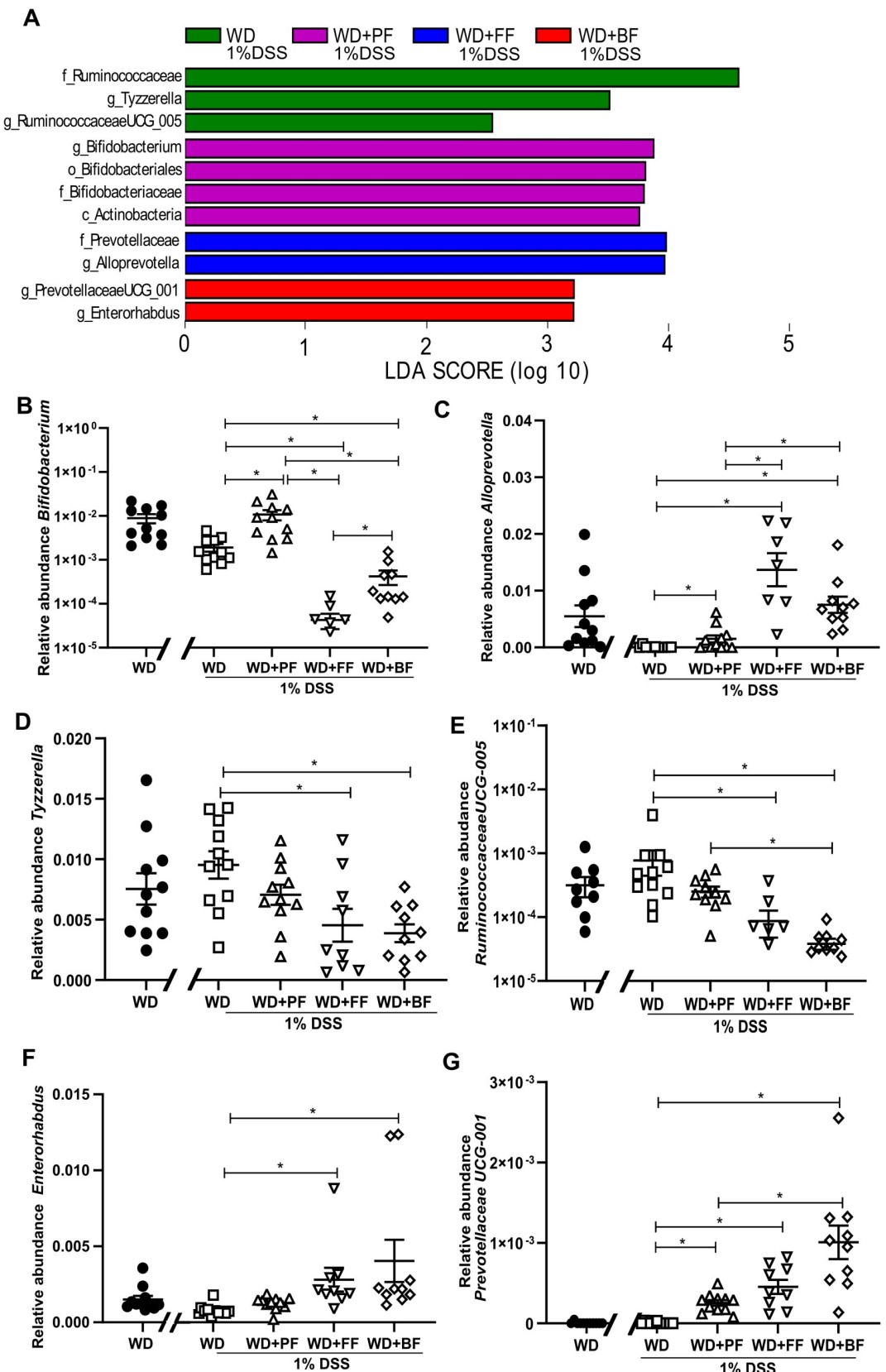

**Fig 6. Linear discriminant analysis (LDA) effect size (LEfSe) analysis of fecal microbiota changes following consumption of a high-fat diet (WD) with various faba-bean fractions (PF, FF, BF) during dextran sodium sulfate (DSS) treatment.** Histogram (**6A**) shows the LDA scores of the taxa (genera -g, and family -f) with the greatest differences between the groups. The relative abundance of the genera from histogram are presented in figures (**6B**-**6G**). Values are expressed as means ± SEM, n = 9–11. For panels B, C, D, E, and G, one-way ANOVA with Tukey's multiple comparison was applied, whereas for panel F, the nonparametric test Kruskal-Wallis test with Dunn's multiple comparisons test. * means significantly different ($P<0.05$). Control mice indicated by black circles in the figures.

ideally mimicking a mild colitis phenotype consistent with low-grade colonic inflammation [46–48]. DSS treatments were combined with a commonly assumed unhealthy WD rich in fat and sucrose and low in dietary fibers, which may create an additional stress to the intestine [49]. We hypothesized that protein and/or fiber fraction(s) from faba beans would alleviate the effects of an induced low-grade colonic inflammation imposed by the 1% DSS treatment in WD fed mice.

Prior to inducing low grade inflammation with 1% DSS we observed that mice in all groups fed WD were healthy and tolerated the faba bean fractions well. Intriguingly mice consuming the WD+PF diet gained more weight and had higher food and energy intake compared to all the other diet groups ($P<0.05$). This could simply be a result of a more palatable diet provided by the proteins in the faba beans, of unknown reasons. It may also be explained by a reduced amount in casein in the WD+PF diet independent of palatability. In the WD+PF diet we exchanged 30% of casein with proteins from faba beans, which could have relevance for weight gain. Indeed, a mouse study from 2016 showed that casein, compared to many other protein sources had a significant weight reducing effect in mice fed a high fat diet [50]. When assessing glucose regulation, we did not observe differences between the groups despite weight differences. This is line with Lamming and coworkers who noticed that consumption of plant proteins for a short term period did not affect glucose homeostasis in C57BL/6J mice [51].

When mice were challenged with 1% DSS, we observed that the effect on inflammatory markers in the colon were overall low, and we speculate that the dose used was borderline to induce a robust inflammation in the colon judged from the variable and low induction of proinflammatory gene expression. Nevertheless, we found that 1% DSS led to a significant shortening of colon lenght in all the groups, a robustly higher level of fecal LCN2 and a modest elevation of LBP in plasma, all indicative of a low grade inflammatory state in the colon following the 1% DSS treatment. When assessing the effects of the different faba bean fractions in the DSS exposed enviroment, our results do not support that they were able to mitigate any of the clinical markers of low grade inflammation.

Although we found variable expression of proinflammatory genes, we in fact observed that some of the genes were more highly expressed in the WD+faba bean fractions than mice fed the pure WD following DSS treatment. This was particularly apparent for *Nox2*, a gene expressed in activated macrophages and associated to the respiratory burst during inflammation. We therefore speculate that, in the current conditions, faba beans could potentially increase the DSS effects and impact colon health adversely. This assumption is partially in agreement with an earlier mouse study, in which bean diets exerted both beneficial and adverse effects in colons exposed to 2% DSS when mice were fed bean diets together with DSS [52]. In other relevant studies, legume containing diets were instead swithed to a common basal diet just before DSS exposure and kept on this diet during DSS challenge [13–15, 52–56]. A rationale for such approach was to mimic IBD patients consumption pattern when they experience intestinal problems (thus abstain from legumes and other fermentable sources). The impact of the faba bean diets employed in our study could therefore be impacted by the order of feeding and DSS. Although our results contrast other findings with respect to

legumes' impact on inflammation, a study by Bibi and colleagues showed that high fat fed mice supplemented with peas displayed no difference in colitis symptoms between HFD and HFD+peas during DSS challenge in mice exposed to DSS [57]. Nevertheless, they found that the recovery phase became shorter in the HFD+pea mice. In our case we terminated the experiment at day six of DSS treatment, thus we do not have results from any recovery phase and therefore we are not able to refer to potential contribution [57].

Regarding microbiota in fecal samples of 1% DSS treated mice, we observed a trend of increased alpha-diversity in all faba bean fraction fed mice compared to the WD+DSS mice, whereas the beta-diversity was only clearly different between control mice and those exposed to DSS. These results, which are in line with previous observations using higher doses of DSS (2–5%) [58, 59], indicate that even a low dose of DSS is capable of shifting the bacterial community bacteria structure but not the bacterial species richness. According to Singh *et al.*, plant-protein diets in humans are linked with higher microbiota richness and diversity than animal-protein diets [60]. One potential reason is that dietary fiber [61] and phenolic compounds [62] present in legumes, which are metabolised by intestinal microbiota causing shifts in gut bacterial populations. At phylum level, the relative abundance of Proteobacteria and Verucomicrobia was higher in all mice exposed to 1% DSS, which is in agreement with other DSS administered studies [63]. In mice fed solely a WD, these two phyla had a tendency to be more abundant compared to mice fed a WD supplemented with faba bea fractions. Increased abundance of Proteobacteria is common in IBD and is regarded as indicator of an inflammatory phenotype [64]. The phylum Verrucomicrobia, which has only one representative in the human and mouse gut (*Akkermansia muciniphila*), is characterized by mucin degrading properties. The role of genus *Akkermansia* is conflicting since some studies support its beneficial contribution to intestinal homeostasis [65], whereas other studies indicate that they exacerbate intestinal inflammation [66]. Furthermore, *Akkermansia muciniphila* in mice is linked with both a positive effect in mice fed a high fat diet [67] and harmful effect such as enhanced colitis [68].

Moreover, WD and WD+PF after DSS treatment had significantly higher relative abundance of the phylum Actinobacteria when compared to WD+FF. In addition, we noticed that the F/B ratio was high in WD-fed mice not treated with 1% DSS, whereas all DSS treated mice regardless of diet, revealed a low F/B ratio. The former condition is commonly associated to obesity [69, 70] and the latter with IBDs [71, 72]. Further Lefse analysis, provided detailed information about lower taxonomic groups. Characteristically, the genus Alloprevotella, a carbohydrate fermenting bacteria had high relative abundance in the mice following a WD+FF whereas the genus *Bifidobacterium*, which belongs to probiotic bacteria was highly present in the WD+PF-fed mice. Initially, Hayakawa showed that purifiried raffinose oligosaacharides family promotes Bifidobacteria growth [73]. Our protein fraction diet contains higher amounts of starch and non starch-digestible carbohydrates (raffinose, stachyose, verbascose), whereas the fiber fraction diet is mainly rich in cellulose and other indigestible fibers. Finally, it is important to note that we cannot rule out that differences between the faba-bean fraction groups could have happened already before the DSS treatment. The protein fraction, in particular, led to an increase in weight prior to DSS due to increased energy intake. Optimally, future studies should include non-DSS groups receiving similar diets. Alternatively, the microbiota composition should also be characterized at start of DSS treatment to account for differential impact of the diets prior to DSS.

In conclusion, we assessed whether a high fat Western diet supplemented with faba bean fractions reduced vulnerability towards colonic inflammation induced by a low DSS dose. Herein, our results suggest that although Faba bean fractions could modulate microbiota, they were not able to influence colonic inflammation induced by DSS.

## Supporting information

**S1 Fig. Histology images.** Representative images of colon tissue from mice exposed to 1% DSS. Colon sections (7μm) were stained with hematoxylin and eosin. BF, both fractions; DSS, dextran sodium sulfate; FF, fiber fraction; PF, protein fraction; WD, western diet.
(PDF)

**S1 Table.** a. Composition of food pellets. b. Content of cooked faba bean fractions.
(PDF)

**S2 Table. Criteria for DAI and scoring way for assessing during exposure to 1% DSS.**
(PDF)

**S3 Table. Reaction mixture for cDNA synthesis using the iScript cDNA Synthesis Kit.**
(PDF)

**S4 Table. Temperature program used for cDNA synthesis.**
(PDF)

**S5 Table. Primer sequences for RT-qPCR and melting temperatures.**
(PDF)

**S6 Table. Reaction mixture for amplicon PCR during library preparation for gene sequencing of 16S rRNA.**
(PDF)

**S7 Table. Temperature cycles for amplicon PCR during library preparation for gene sequencing of 16S rRNA.**
(PDF)

**S8 Table. Reaction mixture for index PCR during library preparation for gene sequencing of 16S rRNA.**
(PDF)

**S9 Table. Temperature cycles for index PCR during library preparation for gene sequencing of 16S rRNA.**
(PDF)

**S10 Table. Primers modified with Illumina adapters used for index PCR during library preparation for gene sequencing of 16S rRNA.**
(PDF)

## Acknowledgments

We thank Knut Rudi, Morten Nilsen and Ida Ormaasen for advice and lab facilities for microbiota sequencing, Lars Fredrik Moen, Silje Harvei and Henriette Arnesen for contributing with mouse experiments. A special thanks to Svein Halvor Knutsen, Stefan Sahlstrøm and Catia Saldanha do Carmo at NOFIMA AS for providing the faba bean fractions.

## Author Contributions

**Conceptualization:** Harald Carlsen.

**Data curation:** Dimitrios Papoutsis, Anne Mari Herfindal.

**Formal analysis:** Dimitrios Papoutsis, Sérgio Domingos Cardoso Rocha.

**Funding acquisition:** Harald Carlsen.

**Investigation:** Dimitrios Papoutsis.

**Methodology:** Dimitrios Papoutsis, Sérgio Domingos Cardoso Rocha, Harald Carlsen.

**Project administration:** Harald Carlsen.

**Supervision:** Siv Kjølsrud Bøhn, Harald Carlsen.

**Writing – original draft:** Dimitrios Papoutsis.

**Writing – review & editing:** Sérgio Domingos Cardoso Rocha, Anne Mari Herfindal, Siv Kjølsrud Bøhn, Harald Carlsen.

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
