## [Decision Letter · Decision Letter 0]

24 May 2022

PONE-D-22-03545Intestinal effect of faba bean fractions in WD-fed mice treated with low dose of DSSPLOS ONE

Dear Dr. Carlsen,

Thank you for submitting your manuscript to PLOS ONE. After careful consideration, we feel that it has merit but does not fully meet PLOS ONE’s publication criteria as it currently stands. Therefore, we invite you to submit a revised version of the manuscript that addresses the points raised during the review process. We have received a thoughtful review of your manuscript. I agree with the comments from the reviewer.  In order to not delay the review process any further, I would like to provide you with the opportunity to revise your manuscript based on this one review.

We look forward to receiving your revised manuscript.

Kind regards,

Michael W. Greene, Ph.D.

Academic Editor

PLOS ONE

Journal Requirements:

2. Please further describe the source of the Faba beans used in this study.

In your cover letter, please note whether your blot/gel image data are in Supporting Information or posted at a public data repository, provide the repository URL if relevant, and provide specific details as to which raw blot/gel images, if any, are not available. Email us at plosone@plos.org if you have any questions

Reviewers' comments:

Reviewer's Responses to Questions

**Comments to the Author**

1. Is the manuscript technically sound, and do the data support the conclusions?

Reviewer #1: Partly

2. Has the statistical analysis been performed appropriately and rigorously? 

Reviewer #1: Yes

3. Have the authors made all data underlying the findings in their manuscript fully available?

Reviewer #1: Yes

4. Is the manuscript presented in an intelligible fashion and written in standard English?

Reviewer #1: Yes

5. Review Comments to the Author

Reviewer #1: Ms. ID PONE-D-22-03545: " Intestinal effect of faba bean fractions in WD-fed mice treated with low dose of DSS"

General comments:

The aim of the present study was to determine the ability aba bean fractions to reduce low-grade inflammation induced by a Western diet (High Fat diet) and the addition of DSS during the last six days before the end of experiment, and to modulate the intestinal microbiota of male C57BL/6JRj mice after 8 weeks. Wester diet (WD) and Protein Fraction (PF) diet increased the weight gain as compared to WD and WD+ Fiber Fraction (FF) and no signification difference was observed for glucose regulation. Protein and fiber fraction from faba beans did not alleviate the effect of low-grade colonic inflammation induced by the 1% DSS treatment. Faba bean fractions seems able to modulate gut microbiota. Therefore, the authors concluded that the replacement of protein and or fiber in WD with htose of faba bean fraction had marginal effects on colonic and inflammation parameters. The role of faba bean fractions as gut microbiota modulator was not clearly demonstrated although fiber fraction seems to modulate the relative abundance of Alloprevotella and Prevotellaceae.

This study is built on basic analysis and mice model that provides preliminary information on the effects of faba bean fractions on alleviation of low-grade inflammation induced by DSS and their impact on modulation of gut microbiota.

Specific comments:

Abstract

Line 46: but modulated intestinal microbiota. Comments: The results are not as convincing, it seems that the fiber fraction can modulate members of the prevotella family. However, this same fraction reduces the relative abodance of bifidobacteria. It would be more prudent to point out that the faba fractions also had marginal effects on intestinal microbiota.

Introduction

Lines 90-94 : Mice fed a western diet may develop low-grade inflammation. This low-grade inflammation can also be simulated by inducing it with a DSS intervention. Here, the authors chose to do both. A rationale for this model would be needed or supported by references (see lines 354-356, no reference).

Results

Line 330-331: Please specify higher relative abundance levels instead of higher levels. In addition, these results should be discussed as higher relative abundance levels of Proteobacteria and mucin-degrading bacteria (Verrucomicrobia) are observed in experiments that sum up DSS-induced chronic inflammations, suggesting a dysbiosis. The fractions studied would therefore not have an effect to reduce this dysbiosis.

Discussion

Lines 402-407: The comment on the higher relative abundance of proteobacteria and mucin-degrading bacteria observed in DSS treated WD mice should be discussed.

6. PLOS authors have the option to publish the peer review history of their article (what does this mean?). If published, this will include your full peer review and any attached files.

Reviewer #1: No

---

## [Author Response · Author response to Decision Letter 0]

27 Jun 2022

Response to reviewers have been uploaded in a separate document.

---

## [Decision Letter · Decision Letter 1]

18 Jul 2022

Intestinal effect of faba bean fractions in WD-fed mice treated with low dose of DSS

PONE-D-22-03545R1

Dear Dr. Carlsen,

We’re pleased to inform you that your manuscript has been judged scientifically suitable for publication and will be formally accepted for publication once it meets all outstanding technical requirements.

Kind regards,

Michael W. Greene, Ph.D.

Academic Editor

PLOS ONE

Additional Editor Comments (optional):

Reviewers' comments:

Reviewer's Responses to Questions

**Comments to the Author**

1. If the authors have adequately addressed your comments raised in a previous round of review and you feel that this manuscript is now acceptable for publication, you may indicate that here to bypass the “Comments to the Author” section, enter your conflict of interest statement in the “Confidential to Editor” section, and submit your "Accept" recommendation.

Reviewer #1: All comments have been addressed

2. Is the manuscript technically sound, and do the data support the conclusions?

Reviewer #1: Yes

3. Has the statistical analysis been performed appropriately and rigorously? 

Reviewer #1: Yes

4. Have the authors made all data underlying the findings in their manuscript fully available?

Reviewer #1: Yes

5. Is the manuscript presented in an intelligible fashion and written in standard English?

Reviewer #1: Yes

6. Review Comments to the Author

Reviewer #1: All comments have been addressed - no additional comments, this manuscript is technically sound, well-written

7. PLOS authors have the option to publish the peer review history of their article (what does this mean?). If published, this will include your full peer review and any attached files.

Reviewer #1: No

---

## [Editor Report · Acceptance letter]

30 Jul 2022

PONE-D-22-03545R1 

Intestinal effect of faba bean fractions in WD-fed mice treated with low dose of DSS 

Dear Dr. Carlsen:

I'm pleased to inform you that your manuscript has been deemed suitable for publication in PLOS ONE. Congratulations! Your manuscript is now with our production department. 

Kind regards, 

on behalf of

Dr. Michael W. Greene 

Academic Editor

PLOS ONE